# Direct Runge-Kutta Discretization Achieves Acceleration

**Jingzhao Zhang**
LIDS
Massachusetts Institute of Technology
Cambridge, MA, 02139
jzhzhang@mit.edu

**Aryan Mokhtari**
LIDS
Massachusetts Institute of Technology
Cambridge, MA, 02139
aryanm@mit.edu

**Suvrit Sra**
LIDS, IDSS
Massachusetts Institute of Technology
Cambridge, MA, 02139
suvrit@mit.edu

**Ali Jadbabaie**
LIDS, IDSS
Massachusetts Institute of Technology
Cambridge, MA, 02139
jadbabai@mit.edu

## Abstract

We study gradient-based optimization methods obtained by directly discretizing a second-order ordinary differential equation (ODE) related to the continuous limit of Nesterov's accelerated gradient method. When the function is smooth enough, we show that acceleration can be achieved by a stable discretization of this ODE using standard Runge-Kutta integrators. Specifically, we prove that under Lipschitz-gradient, convexity and order-$(s + 2)$ differentiability assumptions, the sequence of iterates generated by discretizing the proposed second-order ODE converges to the optimal solution at a rate of $\mathcal{O}(N^{-2\frac{s}{s+1}})$, where $s$ is the order of the Runge-Kutta numerical integrator. Furthermore, we introduce a new local flatness condition on the objective, under which rates even faster than $\mathcal{O}(N^{-2})$ can be achieved with low-order integrators and only gradient information. Notably, this flatness condition is satisfied by several standard loss functions used in machine learning. We provide numerical experiments that verify the theoretical rates predicted by our results.

## 1 Introduction

We study accelerated first-order optimization algorithms for the problem

$$\min_{x \in \mathbb{R}^d} \quad f(x), \tag{1}$$

where $f$ is convex and sufficiently smooth. A classical method for solving (1) is gradient descent (GD), which displays a sub-optimal convergence rate of $\mathcal{O}(N^{-1})$—i.e., the gap $f(x_N) - f(x^*)$ between GD and the optimal value $f(x^*)$ decreases to zero at the rate of $\mathcal{O}(N^{-1})$. Nesterov's seminal accelerated gradient method [19] matches the oracle lower bound of $O(N^{-2})$ [18], and is thus a central result in the theory of convex optimization.

However, ever since its introduction, acceleration has remained somewhat mysterious, especially because Nesterov's original derivation relies on elegant but unintuitive algebraic arguments. This lack of understanding has spurred a variety of recent attempts to uncover the rationale behind the phenomenon of acceleration [1, 9, 11, 13, 16, 21].

We pursue instead an approach to NAG (and accelerated methods in general) via a continuous-time perspective. This view was recently studied by Su et al. [23], who showed that the continuous limit

of NAG is a second order ODE describing a physical system with vanishing friction; Wibisono et al. [26] generalized this idea and proposed a class of ODEs by minimizing Bregman Lagrangians.

Although these works succeed in providing a richer understanding of Nesterov's scheme via its continuous time ODE, they fail to provide a general discretization procedure that generates *provably convergent* accelerated methods. In contrast, we introduce a second-order ODE that generates an accelerated first-order method for smooth functions if we simply discretize it using *any* Runge-Kutta numerical integrator and choose a suitable step size.

## 1.1 Summary of results

Assuming that the objective function is convex and sufficiently smooth, we establish the following:

- We propose a second-order ODE, and show that the sequence of iterates generated by discretizing using a Runge-Kutta integrator converges to the optimal solution at the rate $\mathcal{O}(N^{\frac{-2s}{s+1}})$, where $s$ is the order of the integrator. By using a more precise numerical integrator, (i.e., a larger $s$), this rate approaches the optimal rate $\mathcal{O}(N^{-2})$.

- We introduce a new local flatness condition for the objective function (Assumption 1), under which Runge-Kutta discretization obtains convergence rates even faster than $\mathcal{O}(N^{-2})$, without requiring high-order integrators. In particular, we show that if the objective is locally flat around a minimum, by using *only gradient information* we can obtain a convergence rate of $\mathcal{O}(N^{-p})$, where $p$ quantifies the degree of local flatness. Acceleration due to local flatness may seem counterintuitive at first, but our analysis reveals why it helps.

To the best of our knowledge, this work presents the first direct[1] discretization of an ODE that yields accelerated gradient methods. Unlike Betancourt et al. [7] who study symplecticity and consider variational integrators, and Scieur et al. [22] who study consistency of integrators, we focus on the order of integrators (see §2.1). We argue that the stability inherent to the ODE and order conditions on the integrators suffice to achieve acceleration.

## 1.2 Additional related work

Several works [2, 3, 5, 8] have studied the asymptotic behavior of solutions to dissipative dynamical systems. However, these works retain a theoretical focus as they remain in the continuous time domain and do not discuss the key issue, namely, stability of discretization. Other works such as [15], study the counterpart of Su et al. [23]'s work for mirror descent algorithms and achieve acceleration via Nesterov's technique. Diakonikolas and Orecchia [10] proposes a framework to analyze the first order mirror descent algorithms by studying ODEs derived from duality gaps. Also, Raginsky and Bouvrie [20] obtain nonasymptotic rates for continuous time mirror descent in a stochastic setting.

A textbook treatment of numerical integration is given in [12]; some of our proofs build on material from Chapters 3 and 9. [14] and [25] also provide nice introductions to numerical analysis.

## 2 Problem setup and background

Throughout the paper we assume that the objective $f$ is convex and sufficiently smooth. Our key result rests on two key assumptions introduced below. The first assumption is a *local* flatness condition on $f$ around a minimum; our second assumption requires $f$ to have bounded higher order derivatives. These assumptions are sufficient to achieve acceleration simply by discretizing suitable ODEs without either resorting to reverse engineering to obtain discretizations or resorting to other more involved integration mechanisms.

We will require our assumptions to hold on a suitable subset of $\mathbb{R}^d$. Let $x_0$ be the initial point to our proposed iterative algorithm. First consider the sublevel set

$$\mathcal{S} := \{x \in \mathbb{R}^d \mid f(x) \leq \exp(1)((f(x_0) - f(x^*) + \|x_0 - x^*\|^2) + 1\}, \qquad (2)$$

where $x^*$ is a minimum of (1). Later we will show that the sequence of iterates obtained from discretizing a suitable ODE never escapes this sublevel set. Thus, the assumptions that we introduce

need to hold only within a subset of $\mathbb{R}^d$. Let this subset be defined as

$$\mathcal{A} := \{x \in \mathbb{R}^d \mid \exists x' \in \mathcal{S}, \|x - x'\| \leq 1\}, \tag{3}$$

that is, the set of points at unit distance to the initial sublevel set (2). The choice of unit distance is arbitrary, and one can scale that to any desired constant.

**Assumption 1.** *There exists an integer $p \geq 2$ and a positive constant $L$ such that for any point $x \in \mathcal{A}$, and for all indices $i \in \{1, ..., p-1\}$, we have the lower-bound*

$$f(x) - f(x^*) \geq \tfrac{1}{L} \|\nabla^{(i)} f(x)\|^{\frac{p}{p-i}}, \tag{4}$$

*where $x^*$ minimizes $f$ and $\|\nabla^{(i)} f(x)\|$ denotes the operator norm of the tensor $\nabla^{(i)} f(x)$.*

Assumption 1 bounds high order derivatives by function suboptimality, so that these derivatives vanish as the suboptimality converges to 0. Thus, it quantifies the flatness of the objective around a minimum.[2] When $p = 2$, Assumption 1 is slightly weaker than the usual Lipschitz-continuity of gradients (see Example 1) typically assumed in the analysis of first-order methods, including NAG. If we further know that the objectives Taylor expansion around an optimum does not have low order terms, p would be the degree of the first nonzero term.

**Example 1.** *Let $f$ be convex with $\frac{L}{2}$-Lipschitz continuous gradients, i.e., $\|\nabla f(x) - \nabla f(y)\| \leq \frac{L}{2} \|x - y\|$. Then, for any $x, y \in \mathbb{R}^d$ we have*

$$f(x) \geq f(y) + \langle \nabla f(y), x - y \rangle + \tfrac{1}{L} \|\nabla f(x) - \nabla f(y)\|^2.$$

*In particular, for $y = x^*$, an optimum point, we have $\nabla f(y) = 0$, and thus we have $f(x) - f(x^*) \geq \frac{1}{L} \|\nabla f(x)\|^2$, which is nothing but inequality (4) for $p = 2$ and $i = 1$.*

**Example 2.** *Consider the $\ell_p$-norm regression problem: $\min_x f(x) = \|Ax - b\|_p^p$, for even integer $p \geq 2$. If $\exists x^*, Ax^* = b$, then $f$ satisfies inequality (4) for $p$, and $L$ depends on $p$ and the operator norm of $A$.*

Logistic loss satisfies a slightly different version of Assumption 1 because its minimum can be at infinity. We will explain this point in more detail in Section 3.1.

Next, we introduce our second assumption that adds additional restrictions on differentiability and bounds the growth of derivatives.

**Assumption 2.** *There exists an integer $s \geq p$ and a constant $M \geq 0$, such that $f(x)$ is order $(s+2)$ differentiable. Furthermore, for any $x \in \mathcal{A}$, the following operator norm bounds hold:*

$$\|\nabla^{(i)} f(x)\| \leq M, \qquad \text{for } i = p, p+1, \ldots, s, s+1, s+2. \tag{5}$$

When the sublevel sets of $f$ are compact and hence the set $\mathcal{A}$ is also compact; as a result, the bound (5) on high order derivatives is implied by continuity. In addition, an $L_p$ loss of the form $\|Ax - b\|_p^p$ also satisfy (5) with $M = p! \|A\|_2^p$.

## 2.1 Runge-Kutta integrators

Before moving onto our new results (§3) let us briefly recall *explicit* Runge-Kutta (RK) integrators used in our work. For a more in depth discussion please see the textbook [12].

**Definition 1.** *Given a dynamical system $\dot{y} = F(y)$, let the current point be $y_0$ and the step size by $h$. An explicit S stage Runge-Kutta method generates the next step via the following update:*

$$g_i = y_0 + h \sum_{j=1}^{i-1} a_{ij} F(g_j), \qquad \Phi_h(y_0) = y_0 + h \sum_{i=1}^{S} b_i F(g_i), \tag{6}$$

*where $a_{ij}$ and $b_i$ are suitable coefficients defined by the integrator; $\Phi_h(y_0)$ is the estimation of the state after time step $h$, while $g_i$ (for $i = 1, \ldots, S$) are a few neighboring points where the gradient information $F(g_i)$ is evaluated.*

Algorithm 1: Input($f, x_0, p, L, M, s, N$) ▷ Constants $p, L, M$ are the same as in Assumptions

  1: Set the initial state $y_0 = [\vec{0}; x_0; 1] \in \mathbb{R}^{2d+1}$

  2: Set step size h = $C/N^{\frac{1}{s+1}}$                                 ▷ C is determined by $p, L, M, s, x_0$

  3: $x_N \leftarrow$ Order-s-Runge-Kutta-Integrator($F, y_0, N, h$)        ▷ F is defined in equation 12

  4: **return** $x_N$

---

By combining the gradients at several evaluation points, the integrator can achieve higher precision by matching up Taylor expansion coefficients. Let $\varphi_h(y_0)$ be the true solution to the ODE with initial condition $y_0$; we say that an integrator $\Phi_h(y_0)$ has order $s$ if its *discretization error* shrinks as

$$\|\Phi_h(y_0) - \varphi_h(y_0)\| = O(h^{s+1}), \qquad \text{as } h \to 0. \tag{7}$$

In general, RK methods offer a powerful class of numerical integrators, encompassing several basic schemes. The *explicit Euler's* method defined by $\Phi_h(y_0) = y_0 + hF(y_0)$ is an explicit RK method of order 1, while the *midpoint* method $\Phi_h(y_0) = y_0 + hF(y_0 + \frac{h}{2}F(y_0))$ is of order 2. Some high-order RK methods are summarized in [24].

## 3 Main results

In this section, we introduce a second-order ODE and use explicit RK integrators to generate iterates that converge to the optimal solution at a rate faster than $\mathcal{O}(1/t)$ (where $t$ denotes the time variable in the ODE). A central outcome of our result is that, at least for objective functions that are smooth enough, it is not the integrator type that is the key ingredient of acceleration, but a careful analysis of the dynamics with a more powerful Lyapunov function that achieves the desired result. More specifically, we will show that by carefully exploiting boundedness of higher order derivatives, we can achieve both stability and acceleration at the same time.

We start with Nesterov's accelerated gradient (NAG) method that is defined according to the updates

$$x_k = y_{k-1} - h\nabla f(y_{k-1}), \qquad y_k = x_k + \frac{k-1}{k+2}(x_k - x_{k-1}). \tag{8}$$

Su et al. [23] showed that the iteration (8) in the limit is equivalent to the following ODE

$$\ddot{x}(t) + \frac{3}{t}\dot{x}(t) + \nabla f(x(t)) = 0, \qquad \text{where } \dot{x} = \frac{dx}{dt} \tag{9}$$

when one drives the step size $h$ to zero. It can be further shown that in the continuous domain the function value $f(x(t))$ decreases at the rate of $\mathcal{O}(1/t^2)$ along the trajectories of the ODE. This convergence rate can be accelerated to an arbitrary rate in continuous time via time dilation as in [26]. In particular, the solution to

$$\ddot{x}(t) + \frac{p+1}{t}\dot{x}(t) + p^2 t^{p-2}\nabla f(x(t)) = 0, \tag{10}$$

has a convergence rate $\mathcal{O}(1/t^p)$. When $p > 2$, Wibisono et al. [26] proposed rate matching algorithms via utilizing higher order derivatives (e.g., Hessians). In this work, we focus purely on first-order methods and study the stability of discretizing the ODE directly when $p \geq 2$.

Though deriving the ODE from the algorithm is solved, deriving the update of NAG or any other accelerated method by directly discretizing an ODE is not. As stated in [26], explicit Euler discretization of the ODE in (9) may not lead to a stable algorithm. Recently, Betancourt et al. [7] observed empirically that Verlet integration is stable and suggested that the stability relates to the symplectic property of the Verlet integration. However, in our proof, we found that ***the order condition of Verlet integration would suffice to achieve acceleration***. Though symplectic integrators are known to be stable, we weren't able to leverage the symplecticity for the dissipative system (11).

This principal point of departure from previous works underlies Algorithm 1, which solves (1) by discretizing the following ODE with an order-$s$ integrator:

$$\ddot{x}(t) + \frac{2p+1}{t}\dot{x}(t) + p^2 t^{p-2}\nabla f(x(t)) = 0. \tag{11}$$

The solution to (11) exists and is unique when $t > 0$. This claim follows by local Lipschitzness of $f$ and is discussed in more details in Appendix A.2 of [26].

We further highlight that the ODE in (11) can also be written as the dynamical system

$$\dot{y} = F(y) = \begin{bmatrix} -\frac{2p+1}{t}v - p^2 t^{p-2}\nabla f(x) \\ v \\ 1 \end{bmatrix}, \qquad \text{where } y = [v; x; t]. \tag{12}$$

We have augmented the state with time to obtain an autonomous system, which can be readily solved numerically with a Runge-Kutta integrator as in Algorithm 1. To avoid singularity at $t = 0$, Algorithm 1 discretizes the ODE starting from $t = 1$ with initial condition $y(1) = y_0 = [0; x_0; 1]$. The choice of 1 can be replaced by any arbitrary positive constant.

Notice that the ODE in (11) is slightly different from the one in (10); it has a coefficient $\frac{2p+1}{t}$ for $\dot{x}(t)$ instead of $\frac{p+1}{t}$. This modification is crucial for our analysis via Lyapunov functions (more details in Section 4 and Appendix A).

The parameter $p$ in the ODE (11) is set to be the same as the constant in Assumption 1 to achieve the best theoretical upper bound by balancing stability and acceleration. Particularly, the larger $p$ is, the faster the system evolves. Hence, the numerical integrator requires smaller step sizes to stabilize the process, but a smaller step size increases the number of iterations to achieve a target accuracy. This tension is alleviated by Assumption 1. The larger $p$ is, the flatter the function $f$ is around its stationary points. In other words, Assumption 1 implies that as the iterates approach a minimum, the high order derivatives of the function $f$, in addition to the gradient, also converge to zero. Consequently, the trajectory slows down around the optimum and we can stably discretize the process with a large enough step size. This intuition ultimately translates into our main result.

**Theorem 1. (Main Result)** *Consider the second-order ODE in* (11). *Suppose that the function $f$ is convex and Assumptions 1 and 2 are satisfied. Further, let $s$ be the order of the Runge-Kutta integrator used in Algorithm 1, $N$ be the total number of iterations, and $x_0$ be the initial point. Also, let $\mathcal{E}_0 := f(x_0) - f(x^*) + \|x_0 - x^*\|^2 + 1$. Then, there exists a constant $C_1$ such that if we set the step size as $h = C_1 N^{-1/(s+1)}(L + M + 1)^{-1}\mathcal{E}_0^{-1}$, the iterate $x_N$ generated after running Algorithm 1 for $N$ iterations satisfies the inequality*

$$f(x_N) - f(x^*) \le C_2 \mathcal{E}_0 \left[ \frac{(L+M+1)\mathcal{E}_0}{N^{\frac{s}{s+1}}} \right]^p \tag{13}$$

*where the constants $C_1$ and $C_2$ only depend on $s$, $p$, and the Runge-Kutta integrator. $S$ is the number of stage as defined in 1. Since each iteration consumes $S$ gradient, $f(x_N) - f(x^*)$ will converge as $\mathcal{O}(S^{\frac{ps}{s+1}} N^{-\frac{ps}{s+1}})$ with respect to the number of gradient evaluations. Note that for commonly used Runge-Kutta integrators, $S \le 8$.*

The proof of this theorem is quite involved; we provide a sketch in Section 4, deferring the detailed technical steps to the appendix. We do not need to know the constant $C_1$ exactly in order to set the step size $h$. Replacing $C_1$ by any smaller positive constant leads to the same polynomial rate.

Theorem 1 indicates that if the objective has bounded high order derivatives and satisfies the flatness condition in Assumption 1 with $p > 0$, then discretizing the ODE in (11) with a high order integrator results in an algorithm that converges to the optimal solution at a rate that is close to $\mathcal{O}(N^{-p})$. In the following corollaries, we highlight two special instances of Theorem 1.

**Corollary 2.** *If the function $f$ is convex with $L$-Lipschitz gradients and is $4^{th}$ order differentiable, then simulating the ODE* (11) *for $p = 2$ with a numerical integrator of order $s = 2$ for $N$ iterations results in the suboptimality bound*

$$f(x_N) - f(x^*) \le \frac{C_2(f(x_0) - f(x^*) + \|x_0 - x^*\|^2 + 1)^3 (L + M + 1)^2}{N^{4/3}}.$$

Note that higher order differentiability allows one to use a higher order integrator, which leads to the optimal $\mathcal{O}(N^{-2})$ rate in the limit. The next example is based on high order polynomial or $\ell_p$ norm.

**Corollary 3.** *Consider the objective function $f(x) = \|Ax + b\|_4^4$. Assume that $\exists x, s.t. Ax = -b$. Simulating the ODE* (11) *for $p = 4$ with a numerical integrator of order $s = 4$ for $N$ iterations results in the suboptimality bound*

$$f(x_N) - f(x^*) \le \frac{C_2(f(x_0) - f(x^*) + \|x_0 - x^*\|^2 + 1)^5 (L + M + 1)^4}{N^{16/5}}.$$

## 3.1 Logistic loss

Discretizing logistic loss $f(x) = \log(1 + e^{-w^T x})$ does not fit exactly into the setting of Theorem 1 due to nonexistence of $x^*$. This potentially causes two problems. First, Assumption 1 is not well defined. Second, the constant $\mathcal{E}_0$ in Theorem 1 is not well defined. We explain in this section how we can modify our analysis to admit logistic loss by utilizing its structure of high order derivatives.

The first problem can be resolved by replacing $f(x^*)$ by $\inf_{x \in \mathbb{R}^d} f(x)$ in Assumption 1; then, the logistic loss satisfies Assumption 1 with arbitrary integer $p > 0$. To approach the second problem, we replace $x^*$ by $\tilde{x}$ that satisfies the following relaxed inequalities. For some $\epsilon_1, \epsilon_2, \epsilon_3 < 1$ we have

$$\langle x - \tilde{x}, \nabla f(x) \rangle \geq f(x) - f(\tilde{x}) - \epsilon_1, \tag{14}$$

$$f(x) - f(\tilde{x}) \geq \frac{1}{L} \|\nabla^{(i)} f(x)\|^{\frac{p}{p-i}} - \epsilon_2, \qquad f(\tilde{x}) - \inf_{x \in \mathbb{R}^d} f(x) \leq \epsilon_3. \tag{15}$$

As the inequalities are relaxed, there exists a vector $\tilde{x} \in \mathbb{R}^d$ that satisfies the above conditions. If we follow the original proof and balance the additional error terms by picking $\tilde{x}$ carefully, we obtain

**Corollary 4.** *(Informal) If the objective is $f(x) = \log(1 + e^{-w^T x})$, then discretizing the ODE* (11) *with an order s numerical integrator for N iterations with step size $h = \mathcal{O}(N^{-1/(s+1)})$ results in a convergence rate of $\mathcal{O}(S^{p\frac{s}{s+1}} N^{-p\frac{s}{s+1}})$.*

## 4 Proof of Theorem 1

We prove Theorem 1 as follows. First(Proposition 5), we show that the suboptimality $f(x(t)) - f(x^*)$ along the continuous trajectory of the ODE (11) converges to zero sufficiently fast. Second(Proposition 6), we bound the discretization error $\|\Phi_h(y_k) - \varphi_h(y_k)\|$, which measures the distance between the point generated by discretizing the ODE and the true continuous solution. Finally(Proposition 7), a bound on this error along with continuity of the Lyapunov function (16) implies that the suboptimality of the discretized sequence of points also converges to zero quickly.

Central to our proof is the choice of a Lyapunov function used to quantify progress. We propose in particular the Lyapunov function $\mathcal{E} : \mathbb{R}^{2d+1} \to \mathbb{R}_+$ defined as

$$\mathcal{E}([v; x; t]) := \frac{t^2}{4p^2} \|v\|^2 + \left\| x + \frac{t}{2p} v - x^* \right\|^2 + t^p (f(x) - f(x^*)). \tag{16}$$

The Lyapunov function (16) is similar to the ones used by Su et al. [23], Wibisono et al. [26], except for the extra term $\frac{t^2}{4p^2} \|v\|^2$. This term allows us to bound $\|v\|$ by $\mathcal{O}(\frac{\mathcal{E}}{t})$. This dependency is crucial for us to achieve the $O(N^{-2})$ bound(see Lemma 11 for more details).

We begin our analysis with Proposition 5, which shows that the function $\mathcal{E}$ is non-increasing with time, i.e., $\dot{\mathcal{E}}(y) \leq 0$. This monotonicity then implies that both $t^p(f(x) - f(x^*))$ and $\frac{t^2}{4p^2} \|v\|^2$ are bounded above by some constants. The bound on $t^p(f(x) - f(x^*))$ provides a convergence rate of $\mathcal{O}(1/t^p)$ on the sub-optimality $f(x(t)) - f(x^*)$. It further leads to an upper-bound on the derivatives of the function $f(x)$ in conjunction with Assumption 1.

**Proposition 5** (Monotonicity of $\mathcal{E}$). *Consider the vector $y = [v; x; t] \in \mathbb{R}^{2d+1}$ as a trajectory of the dynamical system* (12). *Let the Lyapunov function $\mathcal{E}$ be defined by* (16). *Then, for any trajectory $y = [v; x; t]$, the time derivative $\dot{\mathcal{E}}(y)$ is non-positive and bounded above; more precisely,*

$$\dot{\mathcal{E}}(y) \leq -\frac{t}{p} \|v\|^2. \tag{17}$$

The proof of this proposition follows from convexity and (11); we defer the details to Appendix A.

Next, to bound the Lyapunov function for numerical solutions, we need to bound the distance between points in the discretized and continuous trajectories. As in Section 2.1, for the dynamical system $\dot{y} = F(y)$, let $\Phi_h(y_0)$ denote the solution generated by a numerical integrator starting at point $y_0$ with step size $h$. Similarly, let $\varphi_h(y_0)$ be the corresponding true solution to the ODE. An ideal numerical integrator would satisfy $\Phi_h(y_0) = \varphi_h(y_0)$; however, due to discretization error

there is always a difference between $\Phi_h(y_0)$ and $\varphi_h(y_0)$ determined by the order of the integrator as in (7). Let $\{y_k\}_{i=0}^N$ be the sequence of points generated by the numerical integrator, that is, $y_{k+1} = \Phi_h(y_k)$. In the following proposition, we derive an upper bound on the resulting discretization error $\|\Phi_h(y_k) - \varphi_h(y_k)\|$.

**Proposition 6** (Discretization error). *Let $y_k = [v_k; x_k; t_k]$ be the current state of the dynamical system $\dot{y} = F(y)$ defined in (12). Suppose $x_k \in \mathcal{S}$ defined in (2). If we use a Runge-Kutta integrator of order $s$ to discretize the ODE for a single step with a step size $h$ such that $h \leq \min\{0.2, \frac{1}{(1+\kappa)C(1+\mathcal{E}(y_k))(M+L+1)}\}$, then*

$$\|\Phi_h(y_k) - \varphi_h(y_k)\| \leq C'h^{s+1}(M+L+1)\left[\frac{[(1+\mathcal{E}(y_k))]^{s+1}}{t_k} + h\frac{[(1+\mathcal{E}(y_k))]^{s+2}}{t_k}\right], \quad (18)$$

*where the constants $C$, $\kappa$, and $C'$ only depend on $p$, $s$, and the integrator.*

The proof of Proposition 6 is the most challenging part in proving Theorem 1. Details may be found in Appendix B. The key step is to bound $\|\frac{\partial^{s+1}}{\partial h^{s+1}}[\Phi_h(y_k) - \varphi_h(y_k)]\|$. To do so, we first bound the high order derivative tensor $\|\nabla^{(i)}f\|$ using Assumption 1 and Proposition 5 within a region of radius $R$. By carefully selecting $R$, we can show that for a reasonably small $h$, $\Phi_h(y_k)$ and $\varphi_h(y_k)$ is constrained in the region. Second, we need to compute the high order derivatives of $\dot{y} = F(y)$ as a function of $\nabla^{(i)}f$ which is bounded in the region of radius R. As shown in Appendix E, the expressions for higher derivatives become quite complicated as the order increases. We approach this complexity by using the notation for elementary differentials (see Appendix E) adopted from [12]; we then induct on the order of the derivatives to bound the higher order derivatives. The flatness assumption (Assumption 1) provides bounds on the operator norm of high order derivatives relative to the objective function suboptimality, and hence proves crucial in completing the inductive step.

By the conclusion in Proposition 6 and continuity of the Lyapunov function $\mathcal{E}$, we conclude that the value of $\mathcal{E}$ at a discretized point is close to its continuous counterpart. Using this observation, we expect that the Lyapunov function values for the points generated by the discretized ODE do not increase significantly. We formally prove this key claim in the following proposition.

**Proposition 7.** *Consider the dynamical system $\dot{y} = F(y)$ defined in (12) and the Lyapunov function $\mathcal{E}$ defined in (16). Let $y_0$ be the initial state of the dynamical system and $y_N$ be the final point generated by a Runge-Kutta integrator of order $s$ after $N$ iterations. Further, suppose that Assumptions 1 and 2 are satisfied. Then, there exists a constant $\tilde{C}$ determined by $p, s$ and the numerical integrator, such that if the step size $h$ satsfies $h = \tilde{C}\frac{N^{-1/(s+1)}}{(L+M+1)(e\mathcal{E}(y_0)+1)}$, then we have*

$$\mathcal{E}(y_N) \leq \exp(1)\,\mathcal{E}(y_0) + 1. \quad (19)$$

Please see Appendix C for a proof of this claim.

Proposition 7 shows that the value of the Lyapunov function $\mathcal{E}$ at the point $y_N$ is bounded above by a constant that depends on the initial value $\mathcal{E}(y_0)$. Hence, if the step size $h$ satisfies the required condition in Proposition 7, we can see that

$$f(x_N) - f(x^*) \leq \frac{\mathcal{E}(y_N)}{t_N^p} \leq \frac{e\mathcal{E}(y_0)+1}{(1+Nh)^p}. \quad (20)$$

The first inequality in (20) follows from the definition of the $\mathcal{E}$ (16). Replacing the step size $h$ in (20) by the choice used in Proposition 7 yields

$$f(x_N) - f(x^*) \leq \frac{(L+M+1)^p(e\mathcal{E}(y_0)+1)^{p+1}}{\tilde{C}N^{p\frac{s}{s+1}}}, \quad (21)$$

and the claim of Theorem 1 follows.

*Note:* The dependency of the step size $h$ on the degree of the integrator $s$ suggests that an integrator of higher order allows for larger step size and therefore faster convergence rate.

## 5 Numerical experiments

We perform numerical experiments to verify Theorem 1 and compare ODE direct discretizating (DD) methods described in Algorithm 1 against gradient descent (GD) and Nesterov's accelerated

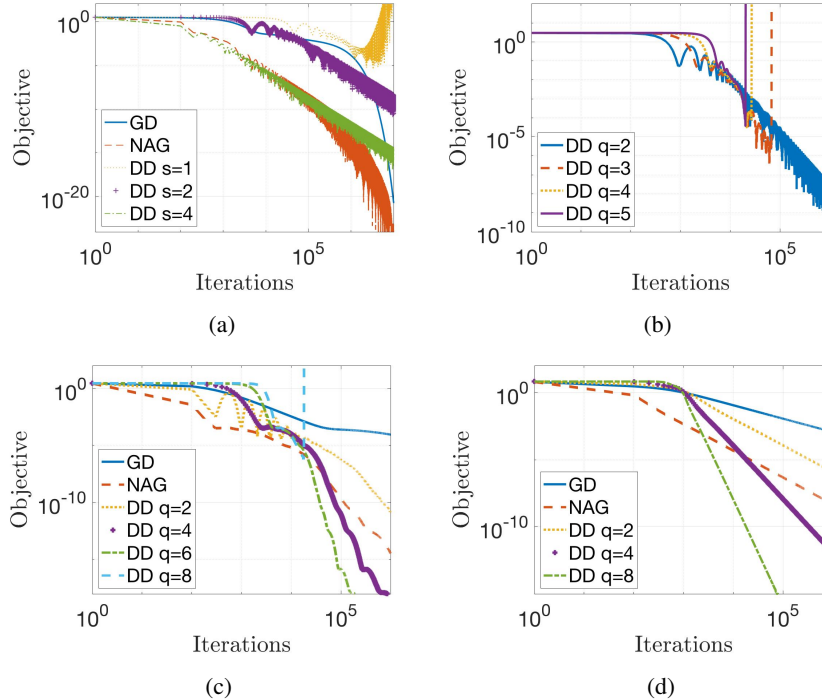

Figure 1: (a) Convergence paths of GD, NAG, and the proposed algorithm with integrators of degree $s = 1$, $s = 2$, and $s = 4$. The objectives is quadratic. (b) Minimizing quadratic objective by discretizing different ODEs (different choices of $q$ in (22)) with the RK44 integrator ($4^{th}$ order). (c/d) Minimizing $L_4$/logistic loss by discretizing different ODEs with a second order integrator.

gradient(NAG) method. All figures in this section are on log-log scale. For each optimization method, we empirically choose the largest step size among $\{10^{-k}|k \in \mathbb{Z}\}$ subject to that the algorithm remains stable in the first 1000 iterations.

In Figure 1a, we generate synthetic linearly separable dataset and fit linear model $Ax = b$. $A$ is entry-wise Gaussian and the feasibility is achieved via increasing data dimension. We then minimize $L_2$ loss $f(x) = \|Ax - b\|_2^2$. In particular, we discretize the ODE (11) for $p = 2$ with integrators of different orders, i.e., $s \in \{1, 2, 4\}$ and compare them against GD and NAD. Observe that GD eventually attains linear rate and NAG achieves local acceleration close to the optimal point as mentioned in [4]. For DD, if we simulate the ODE with an integrator of order $s = 1$, the algorithm is eventually unstable. Using a higher order integrator leads to stable accelerated algorithms.

Throughout this paper, we have assumed that the constant $p$ in (11) is the same as the one in Assumption 1 to attain the best theoretical upper bounds. In Figure 1b, we empirically explore the convergence rate of discretizing the ODE

$$\ddot{x}(t) + \frac{2q+1}{t}\dot{x}(t) + q^2 t^{q-2}\nabla f(x(t)) = 0, \tag{22}$$

when $q \neq p$. We minimize the same $L_2$ loss with different values of $q$ using a fourth order integrator with the same step size. We observe that when $q > 2$, the algorithm diverges eventually. We then discretize ODEs with different parameter $q$ for $L_4$ loss and logistic loss on the same set of data points using a second order RK integrator. As shown in Figure 1c, the objective decreases faster for larger $q$ up to $q = 6$ and diverges when $q = 8$. Given that $L_4$ loss has $p = 4$, this result suggests that our analysis might be conservative. Finally, figure 1d summarizes the experiment result for minimizing logistic loss. We notice that the algorithm is stable even when $q = 8$. This result verifies Corollary 4.

## 6    Discussion

Our paper obtains accelerated gradient methods by directly discretizing second order ODEs (instead of reverse engineering Nesterov-like constructions), yet it does not fully explain acceleration. First,

unlike Nesterov's accelerated gradient method that only requires first order differentiability, our results require the objective function to be $(s + 2)$-times differentiable (where $s$ is the order of the integrator). The precision of numerical integrators only increases with their order when the function is sufficiently differentiable. This property inherently limits our analysis. Second, while we achieve the $\mathcal{O}(N^{-2})$ convergence rate, some of the constants in our bound are loose (e.g., for squared loss and logistic regression they are quadratic in $L$ versus linear in $L$ for NAG). Achieving the optimal dependence on initial errors $f(x_0) - f(x^*)$, the diameter $\|x_0 - x^*\|$, as well as constants $L$ and $M$ requires further investigation.

In addition, we identified a new condition in Assumption 1 that quantifies the *local flatness* of convex functions. At first, this condition may appear counterintuitive, because gradient descent actually converges fast when the objective is *not* flat and the progress slows down if the gradient vanishes close to the minimum. However, when we discretize the ODE, the trajectories with vanishing gradients oscillate slowly, and hence allow stable discretization with large step sizes, which ultimately allows us to achieve acceleration. We think this high-level idea, possibly as embodied by Assumption 1 could be more broadly used in analyzing and designing other optimization methods.

Based on the above two points, this paper contains both positive and negative message for the recent trend in ODE interpretation of optimization methods. On one hand, it shows that with careful analysis, discretizing ODE can preserve some of its trajectories properties. On the other hand, our proof suggests that nontrivial additional conditions might be required to ensure stable discretization. Hence, designing an ODE with nice properties in the continuous domain doesn't guarantee the existence of a practical optimization algorithm.

## Acknowledgement

AJ and SS acknowledge support in part from DARPA FunLoL, DARPA Lagrange; AJ also acknowledges support from an ONR Basic Research Challenge Program, and SS acknowledges support from NSF-IIS-1409802.

## Footnotes

[1]That is, discretize the ODE with known numerical integration schemes without resorting to reverse engineering NAG's updates.

[2]One could view this as an error bound condition that reverses the gradient-based upper bounds on suboptimality stipulated by the Polyak-Łojasiewicz condition [6, 17].

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
