[Supplementary Material · appendix.pdf]

# A Proof of Proposition 5

According to the dynamical system in (12) we can write

$$\dot{x} = v, \qquad \ddot{x} = \dot{v} = -\frac{2p+1}{t}v - p^2 t^{p-2}\nabla f(x). \tag{23}$$

Using these definitions we can show that

$$
\begin{aligned}
\dot{\mathcal{E}} =& \frac{t^2}{4p^2}\langle 2v, \dot{v}\rangle + \frac{2t}{4p^2}\langle v, v\rangle + 2\langle x + \frac{t}{2p}v - x^*, \dot{x} + \frac{\dot{x}}{2p} + \frac{t}{2p}\ddot{x}\rangle + t^p\langle \nabla f(x), \dot{x}\rangle \\
& + pt^{p-1}(f(x) - f(x^*)) \\
=& \frac{2t^2}{4p^2}\langle \dot{x}, \ddot{x} + \frac{2p+1}{t}\dot{x}\rangle - \frac{2t^2}{4p^2}\langle \dot{x}, \frac{2p}{t}\dot{x}\rangle + 2\frac{t}{2p}\langle x + \frac{t}{2p}\dot{x} - x^*, \ddot{x} + \frac{2p+1}{t}\dot{x}\rangle \\
& + t^p\langle \nabla f(x), \dot{x}\rangle + pt^{p-1}(f(x) - f(x^*)) \\
=& \frac{t^2}{2p^2}\langle \dot{x}, -p^2 t^{p-2}\nabla f\rangle - \frac{t}{p}\|\dot{x}\|^2 + \frac{t}{p}\langle x + \frac{t}{2p}\dot{x} - x^*, -p^2 t^{p-2}\nabla f\rangle \\
& + t^p\langle \nabla f(x), \dot{x}\rangle + pt^{p-1}(f(x) - f(x^*)) \\
=& -\frac{t}{p}\|\dot{x}\|^2 + pt^{p-1}(f(x) - f(x^*)) - pt^{p-1}\langle x - x^*, \nabla f\rangle \\
\leq& -\frac{t}{p}\|\dot{x}\|^2. \tag{24}
\end{aligned}
$$

The equalities follows from rearrangement and (11). The last inequality holds due to convexity.

# B Proof of Proposition 6 (Discretization Error)

In this section, we aim to bound the difference between the true solution defined by the ODE and the point generated by the integrator, i.e., $\|\Phi_h(y_c) - \varphi_h(y_c)\|$. Since the integrator has order $s$, the difference $\Delta(h) := \|\Phi_h(y_c) - \varphi_h(y_c)\|$ should be proportional to $h^{s+1}$. Here, we intend to formally derive an upper bound of $\mathcal{O}(h^{s+1})$ on $\Delta(h)$.

We start by introducing some notations. Given a vector $y = [v; x; t] \in \mathbb{R}^{2d+1}$, we define the following projection operators

$$\pi_x(y) = x \in \mathbb{R}^d, \quad \pi_v(y) = v \in \mathbb{R}^d, \quad \pi_t(y) = t \in \mathbb{R}, \quad \pi_{v,x}(y) = \begin{bmatrix} v \\ x \end{bmatrix} \in \mathbb{R}^{2d}. \tag{25}$$

We also define the set $B(x_c, R)$ which is a ball with center $x_c$ and radius $R$ as

$$B(x_c, R) = \{x \in \mathbb{R}^d | \|x - x_c\| \leq R\}, \tag{26}$$

and define the set $U_{R,0.2}(y_c)$ as

$$U_{R,0.2}(y_c) = \{y = [v; x; t] | \|v - v_c\| \leq R, \|x - x_c\| \leq R, |t - t_c| \leq 0.2\}. \tag{27}$$

In the following Lemma, we show that if we start from the point $y_c$ and choose a sufficiently small stepsize, the true solution defined by the ODE $\varphi_h(y_0)$ and the point generated by the integrator $\Phi_h(y_c)$ remain in the set $U_{R,0.2}(y_c)$.

**Lemma 8.** *Let* $y \in U_{R,0.2}(y_c)$ *where* $y_c = [v_c; x_c; t_c]$, $t_c \geq 1$, *and* $R = \frac{1}{t_c}$. *Suppose that* $B(x_c, R) \subseteq \mathcal{A}$ *(defined in (3)) and hence Assumptions 1 and 2 are satisfied. If* $h \leq \min\{0.2, \frac{1}{(1+\kappa)C(\mathcal{E}(y_c)+1)(L+M+1)}\}$, *the true solution defined by the ODE* $\varphi_h(y_0)$ *and the point generated by the integrator* $\Phi_h(y_c)$ *remain in the set* $U_{R,0.2}(y_c)$, *i.e.,*

$$\varphi_h(y_c) \in U_{R,0.2}(y_c), \qquad \Phi_h(y_c) \in U_{R,0.2}(y_c), \tag{28}$$

*where* $\kappa$ *is a constant determined by the Runge-Kutta integrator. In addition, the intermediate points* $g_i$ *defined in Definition 1 also belong to the set* $U_{R,0.2}(y_c)$.

*Proof.* Note that $\forall y \in \mathbb{R}^{2d+1}$, $\|\pi_t F(y)\| = 1$. Clearly when $h \leq 0.2$,

$$\pi_t \varphi_h(y_c) - y_c = h \leq 0.2. \tag{29}$$

Similarly, for any integrator that is at least order 1,

$$\pi_t \Phi_h(y_c) - y_c = h \leq 0.2. \tag{30}$$

Therefore, we only need to focus on bounding the remaining coordinates.

By Lemma 10, we have that when $y \in U_{R,0.2}(y_c)$,

$$\|\pi_{v,x} F(y)\| \leq \frac{C(\mathcal{E}(y_c) + 1)(L + M + 1)}{t_c}. \tag{31}$$

By definition 1,

$$g_i = y_k + h \sum_{j=1}^{i-1} a_{ij} F(g_j) \qquad \Phi_h(y_k) = y_k + h \sum_{i=0}^{s-1} b_i F(g_i).$$

Let $\kappa = \max\{\sum_j |a_{ij}|, \sum |b_i|\}$, we have that when $h \leq \min\{0.2, R/[\kappa \frac{C(\mathcal{E}(y_c)+1)(L+M)}{t_c}]\}$,

$$g_i \in U_{R,0.2}(y_c) \qquad \Phi_h(y_c) \in U_{R,0.2}(y_c). \tag{32}$$

By fundamental theorem of calculus, we have that

$$\varphi_h(y_c) = y_c + \int_0^h F(\varphi_t(y_c))dt \in U_{R,0.2}(y_c). \tag{33}$$

Rearrange and apply Cauchy-Schwarz, we get

$$\|\pi_{v,x}[\varphi_h(y_c) - y_c]\| \leq \int_0^h \|\pi_{v,x} F(\varphi_t(y_c))\|dt \in U_{R,0.2}(y_c). \tag{34}$$

By mean value theorem and proof of contradiction, we can show that when $h \leq \min\{0.2, R/\frac{C(\mathcal{E}(y_c)+1)(L+M)}{t_c}\}$,

$$\int_0^h \|\pi_{v,x} F(\varphi_t(y_c))\|dt \leq R. \tag{35}$$

In particular, if $\int_0^h \|\pi_{v,x} F(\varphi_t(y_c))\|dt \geq R$, then exists $y_1$ and $h_0 < h$ such that $\|y_1 - y_c\| = R$ and $y_1 = y_c + \int_0^{h_0} F(\varphi_t(y_c))dt$. By mean value theorem, this implies that exist $y \in U_{R,0.2}(y_c)$ such that $\|\pi_{v,x} F(y)\| > \frac{C(\mathcal{E}(y_c)+1)(L+M+1)}{t_c}$, which contradicts Lemma 10.

Therefore we proved that

$$\varphi_h(y_c) \in U_{R,0.2}(y_c). \tag{36}$$

$\square$

The result in Lemma 8 shows that $\varphi_h(y_c)$ and $\Phi_h(y_c)$ remain in the set $U_{R,0.2}(y_c)$. In addition, we can bound the operator norm of $\nabla^{(i)} f$ in $B(x_c, R)$ by Lemma 9. Since $\frac{\partial^q \varphi_h(y_c)}{\partial h^q}$ is a function of $\nabla^{(i)} f$, we can show in Lemma 11 that the $(s+1)_{th}$ derivative of $\varphi_h(y_c)$ and $\Phi_h(y_c)$ are bounded above by

$$\left\|\frac{\partial^q \varphi_h(y_c)}{\partial h^q}\right\| \leq \frac{C_0[\mathcal{E}(y_c) + 1]^q (L + M + 1)^q}{t_c}, \tag{37}$$

and

$$\left\|\frac{\partial^q \Phi_h(y_c)}{\partial h^q}\right\| \leq \frac{C_1[1 + \mathcal{E}(y_c)]^q (L + M + 1)^q + C_2 h[1 + \mathcal{E}(y_c)]^{q+1}(L + M + 1)^{p+1}}{t_c}. \tag{38}$$

Since the integrator has order $s$, we can write

$$\frac{\partial^i}{\partial h^i}[\Phi_h(y_k) - \varphi_h(y_k)] = 0 \quad \text{for } i = 1, ..., s. \tag{39}$$

Therefore, the difference between the true solution $\varphi_h(y_c)$ defined by the ODE and the point $\Phi_h(y_c)$ generated by the integrator can be upper bounded by

$$\|\Phi_h(y_c) - \varphi_h(y_c)\| \le \left( \left\| \frac{\partial^{s+1}\varphi_h(y_k)}{\partial h^{s+1}} \right\| + \left\| \frac{\partial^{s+1}\Phi_h(y_k)}{\partial h^{s+1}} \right\| \right) h^{s+1} \tag{40}$$

Replacing the norms on the right hand side of (40) by their upper bounds in (37) and (38) implies that

$$\|\Phi_h(y_c) - \varphi_h(y_c)\| \le h^{s+1} \left[ \frac{(C_0 + C_1)[\mathcal{E}(y_c) + 1]^{s+1}(M + L + 1)^{s+1}}{t_c} \right]$$
$$+ h^{s+2} \left[ \frac{C_2[1 + \mathcal{E}(y_c)]^{s+2}(M + L + 1)^{s+2}}{t_c} \right]. \tag{41}$$

By replacing $y_c = [v_c; x_c; t_c]$ in (41) by $y_k = [v_k; x_k; t_k]$ the claim in (18) follows.

## C   Proof of Proposition 7 (Analysis of discrete Lyapunov functions)

As defined earlier in Section 4, $\Phi_h(y_k)$ is the solution generated by the numerical integrator, and $\varphi_h(y_k)$ is a point on the trajectory of the ODE. $y_c = [\vec{0}; x_c; 1]$ is the initial point of the ODE. Recall that $\{y_k\}_{i=0}^N$ is the sequence of points produced by the numerical integrator, i.e., $y_{k+1} = \Phi_h(y_k)$.

To simplify the notation, we let $E_k = \mathcal{E}(y_k)$, $E_{k+1} = \mathcal{E}(\Phi_h(y_k))$, $\tilde{y} = \varphi_h(y_k) = [\tilde{v}; \tilde{x}; t + h]$, $\hat{y} = \Phi_h(y_k) = [\hat{v}; \hat{x}; t + h]$.

We want to prove by induction on $k = 0, 1, ..., N$ that

$$E_k \le (1 + \frac{1}{N})^k E_0 + \frac{k}{N}. \tag{42}$$

The base case $E_0 \le E_0$ is trivial. Now let's assume by induction that the inequality in (42) holds for $k = j$, i.e.,

$$E_j \le (1 + \frac{1}{N})^j E_0 + \frac{j}{N}. \tag{43}$$

By this assumption, we know that $f(x_k) \le \frac{eE_0+1}{t_k^p} \le eE_0 + 1$ and hence $x_k \in \mathcal{S}$ defined in (2). Note that $R = \frac{1}{t_k} \le 1$. We then have

$$B(x_k, R) \subseteq B(x_k, 1) \in \mathcal{A} \tag{44}$$

for $\mathcal{A}$ defined in (3). By assumption in Proposition 5,

$$h \le 0.2, \qquad h \le \frac{1}{(1 + \kappa)C(eE_0 + 2)(L + M + 1)}. \tag{45}$$

By utilizing the bound on $\|\Phi_h(y_k) - \varphi_h(y_k)\|$ and the continuity of $\mathcal{E}(y)$, we show in Lemma 13 that the discretization error of $\|\mathcal{E}(\hat{y}) - \mathcal{E}(\tilde{y})\|$ is upper bounded by

$$\|\mathcal{E}(\Phi_h(y_k)) - \mathcal{E}(\varphi_h(y_k))\| \tag{46}$$
$$\le C'h^{s+1}[(1 + E_k)^{s+1}(L + M + 1)^{s+1} + h(1 + E_k)^{s+2}(L + M + 1)^{s+2}](E_k + E_{k+1} + 1),$$

under conditions in (44) and (45). $C'$ only depends on $p, s$ and the numerical integrator.

We proceed to prove the inductive step. Start by writing $E_{k+1} = \mathcal{E}(\Phi_h(y_k))$ as

$$\mathcal{E}(\Phi_h(y_k)) = \mathcal{E}(y_k) + \mathcal{E}(\varphi_h(y_k)) - \mathcal{E}(y_k) + \mathcal{E}(\Phi_h(y_k)) - \mathcal{E}(\varphi_h(y_k)). \tag{47}$$

According to Proposition 5, $\mathcal{E}(\varphi_h(y_k)) - \mathcal{E}(y_k) \le 0$. Therefore,

$$E_{k+1} \le E_k + \|\mathcal{E}(\hat{y}) - \mathcal{E}(\tilde{y})\|. \tag{48}$$

Replace the norm $\|\mathcal{E}(\hat{y}) - \mathcal{E}(\tilde{y})\| = \|\mathcal{E}(\Phi_h(y_k)) - \mathcal{E}(\varphi_h(y_k))\|$ by its upper bound (46) to obtain

$$E_{j+1} \le E_j + Ch^{s+1}[(1 + E_j)^{s+1}(L + M + 1)^{s+1} + h(1 + E_j)^{s+2}(L + M + 1)^{s+2}](E_j + E_{j+1} + 1). \tag{49}$$

Before proving the inductive step, we need to ensure that the step size $h$ is sufficiently small. Here, we further add two more $j$-independent conditions on the choice of step size $h$. In particular, we assume that

$$h \leq \frac{1}{eE_0 + 2}, \qquad h^{s+1} \leq \frac{1}{3(1 + C^{-1})C'N(eE_0 + 2)^{s+1}(L + M + 1)^{s+1}}. \qquad (50)$$

Note that since we want show the claim in (42) for $k = 1, \ldots, N$, in inductive assumptions we have that $j \leq N - 1$. Now we proceed to show that if the inequality in (42) holds for $k = j$ it also holds for $k = j + 1$. By setting $k = j$ in (49) we obtain that

$$E_{j+1} \leq E_j + C'h^{s+1}(1+E_j)^{s+1}(L+M+1)^{s+1}[1+h(1+E_j)(L+M+1)](E_j+E_{j+1}+1). \qquad (51)$$

Using the assumption of induction in (43) we can obtain that $E_j \leq eE_0 + 1$ by setting $j = n$ in the right hand side. Using this inequality and the second condition in (45), we can write

$$h \leq \frac{1}{C(eE_0 + 2)(L + M + 1)} \leq \frac{1}{C(E_j + 1)(L + M + 1)} \qquad (52)$$

Using this expression we can simplify (51) to

$$E_{j+1} \leq E_j + (1 + C^{-1})C'h^{s+1}(1 + E_j)^{s+1}(L + M + 1)^{s+1}(E_j + E_{j+1} + 1). \qquad (53)$$

We can further show that

$$(1 + C^{-1})C'h^{s+1}(1 + E_j)^{s+1}(L + M + 1)^{s+1}$$

$$\leq (1 + C^{-1})C'h^{s+1}(2 + eE_0)^{s+1}(L + M + 1)^{s+1} \leq \frac{1}{3N}, \qquad (54)$$

where the first inequality holds since $E_j \leq eE_0 + 1$ and the second inequality holds due to the second condition in (50). Simplifying the right hand side of (53) using the upper bound (54) leads to

$$E_{j+1} \leq E_j + \frac{1}{3N}(E_j + E_{j+1} + 1). \qquad (55)$$

Regroup the terms in (55) to obtain that $E_{j+1}$ is upper bounded by

$$E_{j+1} \leq \left(\frac{1 + \frac{1}{3N}}{1 - \frac{1}{3N}}\right)E_j + \frac{1}{3N - 1} \qquad (56)$$

Now replace $E_j$ by its upper bound in (43) to obtain

$$E_{j+1} \leq \left(\frac{1 + \frac{1}{3N}}{1 - \frac{1}{3N}}\right)\left(\left(1 + \frac{1}{N}\right)^j E_0 + \frac{j}{N}\right) + \frac{1}{3N - 1}$$

$$= \left(\frac{1 + \frac{1}{3N}}{1 - \frac{1}{3N}}\right)\left(1 + \frac{1}{N}\right)^j E_0 + \left(\frac{1 + \frac{1}{3N}}{1 - \frac{1}{3N}}\right)\frac{j}{N} + \frac{1}{3N - 1}$$

$$= \left(\frac{3N + 1}{3N - 1}\right)\left(1 + \frac{1}{N}\right)^j E_0 + \left(\frac{3N + 1}{3N - 1}\right)\frac{j}{N} + \frac{1}{3N - 1}$$

$$\leq \left(1 + \frac{1}{N}\right)^{j+1} E_0 + \left(\frac{3N + 1}{3N - 1}\right)\frac{j}{N} + \frac{1}{3N - 1}, \qquad (57)$$

where the first inequality holds since $\frac{3N+1}{3N-1} \leq \frac{N+1}{N}$ and the last inequality follows from $1 + \frac{2}{3N-1} \geq 1 + \frac{1}{N}$. Further, we can show that

$$\left(\frac{3N + 1}{3N - 1}\right)\frac{j}{N} + \frac{1}{3N - 1} = \left(1 + \frac{2}{3N - 1}\right)\frac{j}{N} + \frac{1}{3N - 1}$$

$$= \frac{j}{N} + \left(\frac{2}{3N - 1}\right)\frac{j}{N} + \frac{1}{3N - 1}$$

$$\leq \frac{j}{N} + \left(\frac{2}{3N - 1}\right)\frac{N - 1}{N} + \frac{1}{3N - 1}$$

$$= \frac{j}{N} + \frac{1}{N}\left(\frac{3N - 2}{3N - 1}\right)$$

$$\leq \frac{j + 1}{N}, \qquad (58)$$

where in the first inequality we use the fact that $j \leq N - 1$. Using the inequalities in (57) and (58) we can conclude that

$$E_{j+1} \leq \left(1 + \frac{1}{N}\right)^{j+1} E_0 + \frac{j+1}{N}, \tag{59}$$

Therefore, the inequality in (42) is also true for $k = j + 1$. The proof is complete by induction and we can write

$$E_N \leq eE_0 + 1. \tag{60}$$

Now if we reconsider the conditions on $h$ in (45) and (50), we can conclude that there exists a constant $\tilde{C}$ that is determined by $p, s$ and the numerical integrator, such that

$$h \leq \tilde{C} \frac{N^{-1/(s+1)}}{(L + M + 1)(eE_0 + 1)}, \tag{61}$$

satisfies all the inequalities in (45) and (50).

## D  Bounding operator norms of derivatives and discretization errors of Lyapunov functions

**Lemma 9.** *Given state $y_c = [v_c; x_c; t_c]$ with $t_c \geq 1$ and the radius $R = \frac{1}{t_c}$, if $B(x_c, R) \subseteq \mathcal{A}$ (defined in (3)) and hence Assumptions 1,2 hold, then for all $y \in U_{R,0.2}(y_c)$ we can write*

$$\|\nabla^{(i)} f(x)\| \leq p(M + L + 1) \frac{\mathcal{E}(y_c) + 1}{t_c^{p-i}}. \tag{62}$$

*Proof.* Based on Assumption 2, we know that

$$\|\nabla^{(p)} f(x)\| \leq M. \tag{63}$$

We further can show that the norm $\|\nabla^{(p-1)} f(x)\|$ is upper hounded by

$$\begin{aligned}
\|\nabla^{(p-1)} f(x)\| &= \|\nabla^{(p-1)} f(x_c) + \nabla^{(p-1)} f(x) - \nabla^{(p-1)} f(x_c)\| \\
&\leq \|\nabla^{(p-1)} f(x_c)\| + \|\nabla^{(p-1)} f(x) - \nabla^{(p-1)} f(x_c)\|
\end{aligned} \tag{64}$$

Using the bound in (63) and the mean value theorem we can show that $\|\nabla^{(p-1)} f(x) - \nabla^{(p-1)} f(x_c)\| \leq M\|x - x_c\| \leq MR$, where the last inequality follows from $y \in U_{R,0.2}(y_c)$. Applying this substitution into (64) implies that

$$\begin{aligned}
\|\nabla^{(p-1)} f(x)\| &\leq \|\nabla^{(p-1)} f(x_c)\| + MR \\
&\leq [L(f(x_c) - f(x^*))]^{\frac{1}{p}} + MR,
\end{aligned} \tag{65}$$

where the first inequality holds due to definition of operator norms and the last inequality holds due to the condition in Assumption 1. By following the same steps one can show that

$$\|\nabla^{(p-2)} f(x)\| \leq [L(f(x_c) - f(x^*))]^{\frac{2}{p}} + R[[L(f(x_c) - f(x^*))]^{\frac{1}{p}} + MR] \tag{66}$$

By iteratively applying this procedure we obtain that if $y = [x; v; t] \in \mathbb{R}^{2d+1}$ belongs to the set $U_{R,0.2}(y_c)$, then we have

$$\|\nabla^{(i)} f(x)\| \leq MR^{p-i} + \sum_{j=i}^{p-1} [L(f(x_c) - f(x^*))]^{\frac{p-j}{p}} R^{j-i}. \tag{67}$$

Notice that since $\frac{p-j}{p} \leq 1$ for $j = 1, \ldots, p - 1$, it follows that we can write $L^{\frac{p-j}{p}} \leq 1 + L$. Moreover, the definition of the Lyapunov function $\mathcal{E}$ in (16) implies that

$$[f(x_c) - f(x^*)]^{\frac{p-j}{p}} \leq \frac{\mathcal{E}(y_c)^{\frac{p-j}{p}}}{t_c^{p-j}} \leq \frac{1 + \mathcal{E}(y_c)}{t_c^{p-j}} \tag{68}$$

where the last inequality follows from the fact that $\mathcal{E}(y_c)^{\frac{p-j}{p}} \leq 1 + \mathcal{E}(y_c)$ for $j = 1, \ldots, p-1$. Therefore, we can simplify the upper bound in (67) by

$$\|\nabla^{(i)} f(x)\| \leq MR^{p-i} + \sum_{j=i}^{p} \frac{(1+L)(1+\mathcal{E}(y_c))}{t_c^{p-j}} R^{j-i}. \tag{69}$$

By replacing the radius $R$ with $1/t_c$ we obtain that

$$\|\nabla^{(i)} f(x)\| \leq \frac{M}{t_c^{p-i}} + \sum_{j=i}^{p} \frac{(1+L)(1+\mathcal{E}(y_c))}{t_c^{p-i}}$$

$$= \frac{M + p(1+L)(1+\mathcal{E}(y_c))}{t_c^{p-i}} \tag{70}$$

As the Lyapunov function $\mathcal{E}(y_c)$ is always non-negative, we can write $M \leq Mp(1 + \mathcal{E}(y_c))$. Applying this substitution into (70) yields

$$\|\nabla^{(i)} f(x)\| \leq \frac{p(L + M + 1)(1 + \mathcal{E}(y_c))}{t_c^{p-i}}, \tag{71}$$

and the claim in (62) follows. $\qquad \square$

**Lemma 10.** *If $B(x_c, R) \subseteq \mathcal{A}$ (defined in (3)) and hence Assumptions 1 and 2 hold, there exists a constant $C$ determined by $p$ such that, $\forall y \in U_{R,0.2}(y_c)$ where $y_c = [v_c; x_c; t_c]$, $t_c \geq 1$ and $R = \frac{1}{t_c}$, we have*

$$\|\pi_{x,v} F(y)\| = \leq \frac{C(\mathcal{E}(y_c) + 1)(L + M + 1)}{t_c}. \tag{72}$$

*Proof.* According to Lemma 9, we can write that

$$\|\nabla f(x)\| \leq p(M + L + 1) \frac{\mathcal{E}(y_c) + 1}{t_c^{p-1}}. \tag{73}$$

Further, the definition of the Lyapunov function in (16) implies that

$$\|v_c\| \leq \frac{2p\mathcal{E}(y_c)^{0.5}}{t_c}. \tag{74}$$

Since $y \in U_{R,0.2}(y_c)$, we have that

$$|t - t_c| \leq 0.2, \qquad \|v - v_c\| \leq R, \qquad \|x - x_c\| \leq R. \tag{75}$$

Further, based on the dynamical system in (12), we can write

$$\|\pi_{x,v} F(y)\| = \left\| \begin{bmatrix} -\frac{2p+1}{t} v - p^2 t^{p-2} \nabla f(x) \\ v \end{bmatrix} \right\|$$

$$\leq \frac{2p+1}{t} \|v\| + \|p^2 t^{p-2} \nabla f(x)\| + \|v\|$$

$$\leq \left( \frac{2p+1}{t} + 1 \right) (\|v_c\| + \|v_c - v\|) + p^2 t^{p-2} \|\nabla f(x)\|, \tag{76}$$

where the first inequality is obtained by using the property of norm, and in the last one we use the triangle inequality. Note that according to (75) we have $t \geq t_c - 0.2$. Since $t_c \geq 1$ it implies that $t \geq 0.8 t_c$. In addition we can also show that $t \leq t_c + 0.2 \leq 1.2 t_c$. Applying these bounds into (76) yields

$$\|\pi_{x,v} F(y)\| \leq \left( \frac{p+1}{0.8 t_c} + 1 \right) (\|v_c\| + \|v_c - v\|) + (1.2)^{p-2} p^2 t_c^{p-2} \|\nabla f(x)\| \tag{77}$$

Replace $\|\nabla f(x)\|$, $\|v_c\|$, and $\|v_c - v\|$ in (77) by their upper bounds in (73), (74), and (75), respectively, to obtain

$$\|\pi_{x,v} F(y)\| \leq \left( \frac{p+1}{0.8 t_c} + 1 \right) \left( \frac{2p\mathcal{E}(y_c)^{0.5}}{t_c} + R \right) + (1.2)^{p-2} p^3 (M + L + 1) \frac{\mathcal{E}(y_c) + 1}{t_c}$$

$$\leq \left( \frac{p+1}{0.8 t_c} + 1 \right) \left( \frac{2p(\mathcal{E}(y_c) + 1) + 1}{t_c} \right) + (1.2)^{p-2} p^3 (M + L + 1) \frac{\mathcal{E}(y_c) + 1}{t_c}, \tag{78}$$

where in the second inequality we replace $R$ by $1/t_c$ and $\mathcal{E}(y_c)^{0.5}$ by its upper bound $\mathcal{E}(y_c) + 1$. Considering that $t_c \geq 1$ and the result in (78) w obtain that there exists a constant $C$ such that

$$\|\pi_{x,v}F(y)\| \leq \frac{C(\mathcal{E}(y_c) + 1)(L + M + 1)}{t_c}, \tag{79}$$

where $C$ only depends on $p$. $\qquad \square$

**Lemma 11.** *Given state $y_c = [v_c, x_c, t_c]$ with $t_c \geq 1$, let $R = \frac{1}{t_c}$. If $B(x_c, R) \subseteq \mathcal{A}$ (defined in (3)) and hence Assumptions 1,2 hold, then when $h \leq \min\{0.2, \frac{1}{(1+\kappa)C(\mathcal{E}(y_c)+1)(L+M+1)}\}$, we have*

$$\left\|\frac{\partial^q \varphi_h(y_c)}{\partial h^q}\right\| \leq \frac{C_0[\mathcal{E}(y_c) + 1]^q(L + M + 1)^q}{t_c}, \tag{80}$$

*and*

$$\left\|\frac{\partial^q \Phi_h(y_c)}{\partial h^q}\right\| \leq \frac{C_1[1 + \mathcal{E}(y_c)]^q(L + M + 1)^q + C_2 h[1 + \mathcal{E}(y_c)]^{q+1}(L + M + 1)^{p+1}}{t_c}, \tag{81}$$

*where $C$ and $\kappa$ are the same constants as in Lemma 10. Further, the constants $C_1, C_2, C_3$ are determined by p, q, and the integrator.*

**Remark 12.** *In the proof below, we reuse variants of symbol $C(e.g.C_1, C_2, \tilde{C})$ to hide constants determined by $p, q$ and the integrator. We recommend readers to focus on the degree of the polynomials in $(L + M + 1), \mathcal{E}(y_c), h, t_c$, and check that the rest can be upper-bounded by variants of symbol $C$. We frequently use two tricks in this section. First, for $a \in (0, 1)$, we can bound*

$$c^a \leq c + 1 \tag{82}$$

*Second, note that given $t_c \geq 1$, for any $n > 0$, there exist constants $C_1, C_2, C_3$ determined by $n$ such that for all $t$ subject to $|t - t_c| \leq 0.2$,*

$$\frac{1}{t^n} \leq \frac{C_1}{t_c^n} \leq C_2 t^n \leq C_3 t_c^n \tag{83}$$

*Proof.* Notice that the system dynamic function $F : \mathbb{R}^{2d+1} \to \mathbb{R}^{2d+1}$ in Equation (12) is a vector valued multivariate function. We denote its $i_{th}$ order derivatives by $\nabla^{(i)}F(y)$, which is a $\underbrace{(2d + 1) \times \ldots \times (2d + 1)}_{i+1 \; times}$ tensor. The tensor is symmetric by continuity and Schwartz theorem. As a shorthand, we use $\nabla^{(i)}F$ to denote $\nabla^{(i)}F(y)$. We know that $y^{(i)} = F^{(i-1)}(y) = \frac{\partial^i y}{\partial t^i}$. Notice that $F^{(i-1)}(y)$ is a vector. As an example, we can write

$$\begin{aligned}
y^{(1)} &= F \\
y^{(2)} &= F^{(1)} = \nabla F(F) \\
y^{(3)} &= F^{(2)} = \nabla^{(2)}F(F, F) + \nabla F(\nabla F(F)).
\end{aligned} \tag{84}$$

The derivative $\nabla^{(i)}F(y)$ can be interpreted as a linear map: $\nabla^{(i)}F : \underbrace{\mathbb{R}^{2d+1} \times \ldots \times \mathbb{R}^{2d+1}}_{i \; times} \to \mathbb{R}^{2d+1}$.

$\nabla^{(2)}F(F_1, F_2)$ maps $F_1, F_2$ to some element in $\mathbb{R}^{2d+1}$. Enumerating the expressions will soon get very complicated. However, we can express them compactly with elementary differentials summarized in Appendix E (see Chapter 3.1 in [12] for details).

First we bound $\nabla^{(i)} F$ by explicitly computing its entries. Let $a(t) = p^2 t^{p-2}$ and $b(t) = \frac{2p+1}{t}$. Based on the definition in (12), we obtain that

$$\frac{\partial^{k+1} F}{\partial v \partial t^k} = \begin{bmatrix} -b^{(k)}(t) I \\ I^{(k)} \\ 0 \end{bmatrix}, \qquad \frac{\partial^k F}{\partial t^k} = \begin{bmatrix} -b^{(k)}(t) v - a^{(k)}(t) \nabla f(x) \\ 0 \\ 0 \end{bmatrix},$$

$$\frac{\partial^{i+k} F}{\partial x^i \partial t^k} = \begin{bmatrix} -a^{(k)}(t) \nabla^{i+1} f(x) \\ 0 \\ 0 \end{bmatrix}, \qquad \frac{\partial^i F}{\partial x^i} = \begin{bmatrix} -a(t) \nabla^{i+1} f(x) \\ 0 \\ 0 \end{bmatrix},$$

$$\frac{\partial^{i+j} F}{\partial v^j \partial x^i} = 0, \qquad \frac{\partial F}{\partial v} = \begin{bmatrix} \frac{2p+1}{t} I \\ I \\ 0 \end{bmatrix}, \qquad \frac{\partial^j F}{\partial v^j} = 0, j \geq 2. \tag{85}$$

$$\tag{86}$$

For any vector $y = [v; x; t] \in U_{R,0.2}(y_c)$, we can show that the norm of $\nabla^{(n)} F$ is upper bounded by

$$\|\nabla^{(n)} F(F_1, F_2, ..., F_n)\| \leq \|a(t) \nabla^{(n+1)} f(x)\| \prod_{i \in [n]} \|\pi_x F_i\|$$

$$+ \|b^{(n)}(t) v + a^{(n)}(t) \nabla f(x)\| \prod_{i \in [n]} \|\pi_t F_i\|$$

$$+ \sum_{k \geq 1}^{n-1} \sum_{\substack{S \subset [n] \\ |S| = k}} \|a^{(k)}(t) \nabla^{(n-k+1)} f(x)\| \left[ \prod_{s \in S} \|\pi_t F_s\| \right] \left[ \prod_{s' \in [n]/S} \|\pi_x F_{s'}\| \right]$$

$$+ \sum_{i \in [n]} \|b^{(n-1)}(t) + 1\| \|\pi_v F_i\| \prod_{j \neq i} \|\pi_t F_j\|. \tag{87}$$

Using the definition of the Lyapunov function $\mathcal{E}$ and the definition of the set $U_{R,0.2}(y_c)$ it can be shown that

$$\|v_c\| \leq \frac{\mathcal{E}(y_c)^{0.5}}{t_c} \leq \frac{\mathcal{E}(y_c) + 1}{t_c}, \quad t_c \geq 1, \quad |t - t_c| \leq 0.2, \quad \|v - v_c\| \leq R. \tag{88}$$

Further, the result in Lemma 9 implies that

$$\|\nabla^{(i)} f(x)\| \leq p(M + L + 1) \frac{\mathcal{E}(y_c) + 1}{t_c^{p-i}}. \tag{89}$$

Substituting the upper bounds in (88) and (89) into (87) implies that for $n = 1, \ldots, p$ we can write

$$\|\nabla^{(n)} F(F_1, F_2, ..., F_n)\|$$

$$\leq C_1 (M + L + 1)[\mathcal{E}(y_c) + 1] t_c^{n-1} \prod_{i \in [n]} \|\pi_x F_i\|$$

$$+ C_2 (M + L + 1) [\mathcal{E}(y_c) + 1] t_c^{-n-1} \prod_{i \in [n]} \|\pi_t F_i\|$$

$$+ C_3 (M + L + 1) \sum_{k \geq 1}^{p-1} [\mathcal{E}(F_c) + 1] t_c^{n-2k-1} \sum_{\substack{S \subset [n] \\ |S| = k}} \left[ \prod_{s \in S} \|\pi_t F_s\| \right] \left[ \prod_{s' \in [n]/S} \|\pi_x F_{s'}\| \right]$$

$$+ C_4 \sum_{i \in [n]} \left[ 1 + \frac{1}{t_c^n} \right] \|\pi_v F_i\| \prod_{j \neq i} \|\pi_t F_j\|, \tag{90}$$

where $C_1, C_2, C_3$, and $C_4$ only depend on $n$ and $p$.

For $n = p, p+1, ..., s$, we can get similar bounds. To do so, not only we use the result in (89), but also we use the bounds guaranteed by Assumption 2. Hence, for $n = p, p+1, ..., s$ it holds

$$\|\nabla^{(n)} F(F_1, F_2, ..., F_n)\|$$
$$\leq C_1 M t_c^{p-2} \prod_{i \in [n]} \|\pi_x F_i\|$$
$$+ C_2(M+L+1)[\mathcal{E}(y_c)+1] t_c^{-n-1} \prod_{i \in [n]} \|\pi_t F_i\|$$
$$+ C_3 \sum_{k \geq 1}^{p-1} (M+L+1)[\mathcal{E}(y_c)+1] t_c^{p-k-2} \sum_{\substack{S \subset [n] \\ |S|=k}} \left[ \prod_{s \in S} \|\pi_t F_s\| \right] \left[ \prod_{s' \in [n]/S} \|\pi_x F_{s'}\| \right]$$
$$+ C_4 \sum_{i \in [n]} \left[ 1 + \frac{1}{t_c^n} \right] \|\pi_v F_i\| \prod_{j \neq i} \|\pi_t F_j\|. \tag{91}$$

Finally we are ready to bound the time derivatives. We first bound the elementary differentials $F(\tau)$ defined in Section E Definition 2. Let $F(\tau) = F(\tau)(y)$ for convenience. We claim that when $|\tau| \leq q$, then $\forall y \in U_{R,0.2}(y_c)$

$$\|\pi_t F(\tau)\| \leq 1, \qquad \|\pi_{v,x} F(\tau)\| \leq C_{|\tau|}(L+M+1)^{|\tau|} \frac{[\mathcal{E}(y_c)+1]^{|\tau|}}{t_c}, \tag{92}$$

where the constant $C_q$ only depends on $p$ and $q$. We use induction to prove the claims in (92). The base case is trivial as we have shown in Lemma 10 that $\|\pi_{x,v} F(\bullet)(y)\| = \|\pi_{x,v} F(y)\| \leq \frac{C(\mathcal{E}(y_c)+1)(L+M)}{t_c}$, and $\|\pi_t F(\bullet)(y)\| = \|\pi_t F(y)\| = 1$. Since the last coordinate grows linearly with rate 1 no matter what $x, v$ are, it can be shown that

$$\pi_t F(\tau)(y) = 0, \forall |\tau| \geq 2. \tag{93}$$

We hence focus on proving the upper bound for the norm $\|\pi_{x,v} F(\tau)(y)\|$ in (92).

Now assume $|\tau| = q$ and it has $m$ subtrees attached to the root, $\tau = [\tau_1, ..., \tau_m]$ with $\sum_{i=1}^m |\tau_i| = q - 1$. When $m \leq p - 1$, by (90) we obtain

$$\|\nabla^{(m)} F(F(\tau_1), ..., F(\tau_m))\|$$
$$\leq C_1[(M+L+1)(\mathcal{E}(y_c)+1)] t_c^{m-1} \prod_{i \in [m]} \|\pi_x F(\tau_i)\|$$
$$+ C_2(M+L+1)[\mathcal{E}(y_c)+1] t_c^{-m-1} \prod_{i \in [m]} \|\pi_t F(\tau_i)\|$$
$$+ C_3 \sum_{k \geq 1}^{m-1} [(M+L+1)(\mathcal{E}(y_c)+1)1] t_c^{m-2k-1} \sum_{\substack{S \subset [m] \\ |S|=k}} \left[ \prod_{s \in S} \|\pi_t F(\tau_s)\| \right] \left[ \prod_{s' \in [m]/S} \|\pi_x F(\tau_{s'})\| \right]$$
$$+ C_4 \sum_{i \in [m]} \left[ 1 + \frac{1}{t_c^n} \right] \|\pi_v F(\tau_i)\| \prod_{j \neq i} \|\pi_t F(\tau_j)\|. \tag{94}$$

Notice that $|\tau_i| \leq q - 1$. By inductive assumption in (92) we can write

$$\|\pi_t F(\tau_i)\| \leq 1 \quad \text{for all } i = 1 \ldots, m \tag{95}$$

$$\prod_{i \in S} \|\pi_{v,x} F(\tau_i)\| \leq C_n(L+M+1)^n \frac{[\mathcal{E}(y_c)+1]^n}{t_c^{|S|}}, \quad \text{where } n = \sum_i |\tau_i|. \tag{96}$$

Apply these substitutions into (94) to and use the inequality $\sum_i |\tau_i| \leq q - 1$ to obtain that

$$\|\nabla^{(m)} F(F(\tau_1), ..., F(\tau_m))\| \leq C_q \frac{[\mathcal{E}(y_c)+1]^q (M+L+1)^q}{t_c}. \tag{97}$$

Hence, since $\|\pi_{x,v}F(\tau)\| \le \|\nabla^{(m)}F(F(\tau_1),...,F(\tau_m))\|$ we obtain that

$$\|\pi_{x,v}F(\tau)\| \le C_q \frac{[\mathcal{E}(y_c)+1]^q(M+L+1)^q}{t_c}. \tag{98}$$

Similarly, for $m \ge p$, by (91) we can write

$$\begin{aligned}
&\|\nabla^{(m)}F(F(\tau_1),...,F(\tau_m))\| \\
&\le C_1 M t_c^{p-2} \prod_{i\in[m]} \|\pi_x F(\tau_i)\| \\
&\quad + C_2(M+L+1)[\mathcal{E}(y_c)+1]t^{-n-1} \prod_{i\in[n]} \|\pi_t F(\tau_i)\| \\
&\quad + C_3 \sum_{k\ge1}^{m-1} [(M+L+1)(\mathcal{E}(y_c)+1)1]t_c^{p-k-2} \sum_{\substack{S\subset[m]\\|S|=k}} \left[\prod_{s\in S} \|\pi_t F(\tau_s)\|\right] \left[\prod_{s'\in[m]/S} \|\pi_x F(\tau_{s'})\|\right] \\
&\quad + C_4 \sum_{i\in[m]} \left[1 + \frac{1}{t_c^n}\right] \|\pi_v F(\tau_i)\| \prod_{j\ne i} \|\pi_t F(\tau_j)\|. \tag{99}
\end{aligned}$$

Plug in the induction assumption in (92) into (99) to obtain

$$\|\pi_{x,v}F(\tau)\| \le \|\nabla^{(m)}F(F(\tau_1),...,F(\tau_m))\| \le C_q \frac{[\mathcal{E}(y_c)+1]^q(M+L+1)^q}{t_c}. \tag{100}$$

Hence, the proof is complete by induction.

Now we proceed to derive an upper bound for higher order time derivatives. By Lemma 14 we can write

$$\|\frac{\partial^q \varphi_h(y_c)}{\partial h^q}\| = \|F^{(q-1)}(\varphi_h(y_c))\| = \|\sum_{|\tau|=q} \alpha(\tau)F(\tau)(\varphi_h(y_c))\|.$$

By Lemma 10, we know that when $h \le \min\{0.2, \frac{1}{(1+\kappa)C(\mathcal{E}(y_c)+1)(M+L)}\}$, $y \in U_{R,0.2}(y_c)$. Therefore, (100) holds. Hence, there exists a constant $C$ determined by $p, q$ such that

$$\|\frac{\partial^q \varphi_h(y_c)}{\partial h^q}\| \le \frac{C[\mathcal{E}(y_c)+1]^q(M+L+1)^q}{t_c}.$$

Similarly by Lemma 15, we have the following equation

$$\frac{\partial^q \Phi_h(y_c)}{\partial h^q} = \sum_{i\le S} b_i[h\frac{\partial^q F(g_i)}{\partial h^q} + q\frac{\partial^{q-1}F(g_i)}{\partial h^q}]$$

Here, $\frac{\partial^q F(g_i)}{\partial h^q}$ has the same recursive tree structure as $F^{(q)}(y)$, except that we need to replace all $F$ in the expression by $\frac{\partial g_i}{\partial h}$ and all $\nabla^{(n)}F(y)$ by $\nabla^{(n)}F(g_i)$. By Definition 1 and Lemma 10, we know that

$$\|\frac{\pi_{x,v}\partial g_i}{\partial h}\| \le \sum_{j\le i-1} |a_{ij}|\frac{C(\mathcal{E}(y_c)+1)(M+L+1)}{t_c}, \qquad \|\frac{\pi_t \partial g_i}{\partial h}\| = |\sum_{j\le i-1} a_{ij}|.$$

We also know by lemma 10 that $\forall i, g_i \in U_{R,0.2}(y_c)$. Hence the bounds for $\|\nabla^{(n)}F(y)\|$ also holds for $\nabla^{(n)}F(g_i)$. Therefore, by the same argument as for bounding $\|\frac{\partial^q \varphi_h(y_c)}{\partial h^q}\|$, we will get same bounds for $\|\frac{\partial^q F(g_i)}{\partial h^q}\|$ up to a constant factor determined by the integrator. Based on this, we conclude that

$$\|\frac{\partial^q \Phi_h(y_c)}{\partial h^q}\| \le \frac{C[(L+M+1)(1+\mathcal{E}(y_c))]^q + C'h[(L+M+1)(1+\mathcal{E}(y_c))]^{(q+1)}}{t_c},$$

where the constants are determined by $p, q$ and the integrator. $\qquad\square$

**Lemma 13.** *Suppose the conditions in Proposition 6 hold. Then, we have that*

$$\|\mathcal{E}(\Phi_h(y_k)) - \mathcal{E}(\varphi_h(y_k))\|$$
$$\leq Ch^{s+1}[(1 + E_k)^{s+1}(L + M + 1)^{s+1} + h(1 + E_k)^{s+2}(L + M + 1)^{s+2}](E_k + E_{k+1} + 1),$$
(101)

*where $C$ only depends on $p, s$ and the numerical integrator.*

*Proof.* Denote $\hat{y} = \Phi_h(y_k), \tilde{y} = \varphi_h(y_k)$. Notice that $\tilde{t} = \hat{t} = t_k + h$. In fact, because we start the simulation at $t_c = 1$ and we require that $h \leq 0.2$, we have

$$\frac{t_k}{\tilde{t}} = \frac{t_k}{t_k + h} \in \left[\frac{5}{6}, 1\right].$$
(102)

Now using the definition of the Lyapunov function $\mathcal{E}$ we can show that

$$\|\mathcal{E}(\hat{y}) - \mathcal{E}(\tilde{y})\| \leq \frac{\tilde{t}^2}{4p^2} \left|\|\tilde{v}\|^2 - \|\hat{v}\|^2\right| + \left|\|\tilde{x} + \frac{\tilde{t}}{2p}\tilde{v} - x^*\|^2 - \|\hat{x} + \frac{\hat{t}}{2p}\hat{v} - x^*\|^2\right| + \tilde{t}^p(|f(\tilde{x}) - f(\hat{x})|)$$

$$\leq \frac{2\tilde{t}^2}{4p^2}(\|\tilde{v} - \hat{v}\|\|\tilde{v} + \hat{v}\|) + \tilde{t}^p(\|\tilde{x} - \hat{x}\|)(\|\nabla f(\tilde{x})\| + \|\nabla f(\hat{x})\|)$$

$$+ 2\left\|\tilde{x} - \hat{x} + \frac{\tilde{t}}{2p}(\tilde{v} - \hat{v})\right\| \left\|\tilde{x} + \frac{\tilde{t}}{2p}\tilde{v} - x^* + \hat{x} + \frac{\hat{t}}{2p}\hat{v} - x^*\right\|,$$
(103)

where to derive the second inequality we used the convexity of the function $f$ which implies

$$\langle y - x, \nabla f(y)\rangle \leq f(x) - f(y) \leq \langle x - y, \nabla f(x)\rangle.$$
(104)

Recall that $E_k = \mathcal{E}(y_k)$, $E_{k+1} = \mathcal{E}(\hat{y}) = \mathcal{E}(\Phi_h(y_k))$, $\tilde{E}_{k+1} = \mathcal{E}(\tilde{y}) = \mathcal{E}(\varphi_h(y_k))$. According to Proposition 5 we know that $\tilde{E}_{k+1} \leq E_k$, and therefore $\tilde{E}_{k+1}$ is upper bounded by $E_k$. Therefore, we can write

$$\|\tilde{v}\| \leq \frac{\sqrt{\tilde{E}_{k+1}}}{\tilde{t}} \leq \frac{\sqrt{E_k}}{\tilde{t}} \leq \frac{E_k + 1}{\tilde{t}}, \qquad \|\hat{v}\| \leq \frac{E_{k+1} + 1}{\hat{t}},$$

$$\left\|\tilde{x} + \frac{\tilde{t}}{2p}\tilde{v} - x^*\right\| \leq \sqrt{E_k} \leq E_k + 1, \qquad \left\|\hat{x} + \frac{\hat{t}}{2p}\hat{v} - x^*\right\| \leq E_{k+1} + 1.$$
(105)

Further, by Assumption 1, we have that

$$\|\nabla f(\tilde{x})\| \leq \frac{L(E_k + 1)}{\tilde{t}^{p-1}}, \qquad \|\nabla f(\hat{x})\| \leq L(f(\hat{x}) - f(x^*))^{\frac{p-1}{p}} \leq L\left(\frac{E_{k+1}}{\hat{t}^p}\right)^{\frac{p-1}{p}} \leq \frac{L(E_{k+1} + 1)}{\hat{t}^{p-1}}.$$
(106)

In addition, by Proposition 6, we know that for some constant C determined by $p, s, L, M$ and the integrator, it holds

$$\max\{\|\tilde{v} - \hat{v}\|, \|\tilde{x} - \hat{x}\|\}$$
$$\leq Ch^{s+1}\left[\frac{[1 + \mathcal{E}(y_k)]^{s+1}(L + M + 1)^{s+1}}{t_k} + h\frac{[1 + \mathcal{E}(y_k)]^{s+2}(L + M + 1)^{s+2}}{t_k}\right].$$
(107)

Define $\mathcal{M} := \left[\frac{[1+\mathcal{E}(y_k)]^{s+1}(L+M+1)^{s+1}}{t_k} + h\frac{[1+\mathcal{E}(y_k)]^{s+2}(L+M+1)^{s+2}}{t_k}\right]$. Use the upper bounds in (105)-(107) and the definition of $\mathcal{M}$ to simplify the right hand side of (103) to

$$\|\mathcal{E}(\hat{y}) - \mathcal{E}(\tilde{y})\| \leq \frac{2\tilde{t}^2}{4p^2}Ch^{s+1}\mathcal{M}\frac{E_k + E_{k+1} + 2}{\tilde{t}} + \tilde{t}^p Ch^{s+1}\mathcal{M}\frac{L(E_{k+1} + E_k + 2)}{\tilde{t}^{p-1}}$$

$$+ 2\left(1 + \frac{t_k}{2p}\right)Ch^{s+1}\mathcal{M}(E_k + E_{k+1} + 2).$$
(108)

Now use the fact that $\frac{t_k}{\tilde{t}}$ is bounded by a constant as shown (102). Further, upper bound all the constants determined by $s, p$ and the numerical integrator, we obtain that

$$\|\mathcal{E}(\hat{y}) - \mathcal{E}(\tilde{y})\|$$
$$\leq C'h^{s+1}[(1 + E_k)^{s+1}(L + M + 1)^{s+1} + h(1 + E_k)^{s+2}(L + M + 1)^{s+2}](E_k + E_{k+1} + 1),$$
(109)

and the claim in (101) follows. □

| $|\tau|$ | $\tau$ | graph | $\alpha(\tau)$ | $F(\tau)$ |
|---|---|---|---|---|
| 1 | $\bullet$ | | 1 | $F$ |
| 2 | $[\bullet]$ | | 1 | $\nabla F(F)$ |
| 3 | $[\bullet, \bullet]$ | | 1 | $\nabla^{(2)}F(F, F)$ |
| 3 | $[[\bullet]]$ | | 1 | $\nabla F(\nabla F(F))$ |
| 4 | $[\bullet, \bullet, \bullet]$ | | 1 | $\nabla^{(3)}F(F, F, F)$ |
| 4 | $[[\bullet], \bullet]$ | | 3 | $\nabla^{(2)}F(\nabla F(F), F)$ |
| 4 | $[[\bullet, \bullet]]$ | | 1 | $\nabla F(\nabla^{(2)}F(F, F))$ |
| 4 | $[[[\bullet]]]$ | | 1 | $\nabla F(\nabla F(\nabla F(F)))$ |

Figure 2: A figure adapted from [12]. Example tree structures and corresponding function derivatives.

# E  Elementary differentials

We briefly summarize some key results on elementary differentials from [12]. For more details, please refer to chapter 3 of the book. Given a dynamical system

$$\dot{y} = F(y)$$

we want to find a convenient way to express and compute its higher order derivatives. To do this, let $\tau$ denote a tree structure as illustrated in Figure 2. $|\tau|$ is the number of nodes in $\tau$. Then we can adopt the following notations as in [12]

**Definition 2.** *For a tree $\tau$, the elementary differential is a mapping $F(\tau) : \mathbb{R}^d \to \mathbb{R}^d$, defined recursively by $F(\bullet)(y) = F(y)$ and*

$$F(\tau) = \nabla^{(m)}F(y)(F(\tau_1)(y), ..., F(\tau_m)(y))$$

*for $\tau = [\tau_1, ..., \tau_m]$. Notice that $\sum_{i=1}^{m} |\tau_i| = |\tau| - 1$.*

Some examples are shown in Figure 2. With this notation, the following results from [12] Chapter 3.1 hold. The proof follows by recursively applying the product rule.

**Lemma 14.** *The qth order derivative of the exact solution to $\dot{y} = F(y)$ is given by*

$$y^{(q)}(t_c) = F^{(q-1)}(y_c) = \sum_{|\tau|=q} \alpha(\tau)F(\tau)(y_c)$$

*for $y(t_c) = y_c$. $\alpha(\tau)$ is a positive integer determined by $\tau$ and counts the number of occurrences of the tree pattern $\tau$.*

The next result is obtained by Leibniz rule. The expression for $\frac{\partial^q F(g_i)}{\partial h^q}$ can be calculated the same way as in Lemma 14.

**Lemma 15.** *For a Runge-Kutta method defined in definition 1, if F is $q_{th}$ differentiable, then*

$$\frac{\partial^q \Phi_h(y_c)}{\partial h^q} = \sum_{i \leq S} b_i [h \frac{\partial^q F(g_i)}{\partial h^q} + q \frac{\partial^{q-1} F(g_i)}{\partial h^q}] \tag{110}$$

*where $\frac{\partial^q F(g_i)}{\partial h^q}$ has the same structure as $F^{(q)}(y)$ in lemma 14, except that we need to replace all F in the expression by $\frac{\partial g_i}{\partial h}$ and all $\nabla^{(n)}F(y)$ by $\nabla^{(n)}F(g_i)$.*