[Reviews · NeurIPS 2018]

Reviewer 1



Brief summary: the authors show that the Runge-Kutta method can be used to discretize a particular second-order ODE (essentially coming from the recent work of Wibisono et al. (2016)), in order to get (near-)optimal convergence rates, provided the loss is convex, smooth, and satisfies two other technical conditions (essentially requiring that the loss be sufficiently flat around its minimizer(s), as well as a uniform bound on the norm of the higher-order derivatives of the loss). A short empirical study is presented to verify the theoretical predictions. Originality: this seems like good work. The originality over the paper of Wibisono et al. (2016) is that here the authors stick to a (family of) first-order method(s) achieving (near-)optimal convergence rates. The proofs used to establish the convergence rate seem to be nonobvious and fairly involved. Clarity: the paper is generally easy to follow. Quality: again, the work here seems good, so I have only a few points to raise. - First of all, I think you should clarify whether the soln to the ODE in Eq 11 exists and is unique … presumably this follows more or less directly from Wibisono et al. (2016)? - As each iteration of Runge-Kutta is more expensive than a single iteration of the Euler method (owing to the \sum_{i=1}^S in Eq 6), the rate is a little "misleading" … can you guys comment on this? - Line 136: the statement in bold seems important, but is never really revisited / explained further … can you clarify in the main text? - Do you have any guidance on how to choose the tuning parameter "p" in practice? - Can you please include the details on how you generated the data A,b for the numerical experiments in Sec 5? (As-is, it seems out of reach to reproduce your experimental results.) Significance: understanding the links between dynamical systems and optimization is an important topic right now. This paper makes a nice contribution to the literature, by showing that ideas from dynamical systems can give rise to optimal algorithms for convex optimization … although it's not totally clear to me how practical the algorithm presented in the paper actually is, at the moment. Update after the author response phase: thanks for responding to my questions. Please do try to include all those points in the revision. I also think that it may not be a bad idea to integrate point 3.1 into the revision, as well.

Reviewer 2



The author analyze the convergence of the RK method applied to a modified version of the "Nesterov Flow" from Su et al. Using a "local flatness" assumption, they show sub-optimal but accelerated rates of convergence in function of 1) the degree of flatness and 2) the order of integration. Clarity ======= The paper is very clear, all the assumptions and modifications to the differential equation of Su et al., as well as the Lyapunov function, are well motivated. Also, the paper is a very good introduction to Runge Kutta methods. Small detail, in section 2.1. : Using "S" for the number of stages and "s" for the order of integration is confusing. In the experiments, by "iteration" you mean the number of gradient calls or the number of "RK" calls? General ======= - Direct acceleration using *any* RK method is a very good point: The analysis is very generic. - The assumption of local flatness up to "s" is a bit strong. Do you think it is possible to weaken that assumption? - In practice, high-order RK methods are quite costly and unstable (because it cancels Taylor coefficients up to "s", but you do not have any control on the coefficient of order "s+1"). Usually, people who uses these methods don't go too far in the number of stages. - Also, high-order runge kutta methods usually needs an incredibly high number of internal stages (for example, RK of order 10 requires 17 stages). Do you takes the number of internal stages into account in you complexity bound? Source: "A Runge-Kutta Method of Order 10", Ernst Hairer. - I have a remark about the choice of the step-size. It looks that the computation of the step "h" is not easy. However, there exist adaptive ways to choose it, based on difference between two RK methods with different orders (ex. RK45, which is implemented in Matlab under the name of ODE45). Could you think this can be used as adaptive stepsize in you experiment? - There exist also other kind of methods: * Linear multi-step methods * Predictor-corrector method * More generally, RK and linear multi steps are instances of "General linear methods" (See the paper "General linear methods" by J. Butcher for instance). Do you think you analysis can be extended to these other standard methods? - I have a remark about the order of the RK method. In your analysis, you said that you can approach the optimal rate of convergence by increasing the order of the method. However, I think design high-order runge kutta is quite painful in practice (for example, RK10 requires to generates, then solve, a system of 1000+ equations). Do you think is required in practice (in particular, when you use adaptive step size)? Decision ======== accept (7/10). I particularly like the fact that *direct* discretisation leads to accelerated rate, but the paper lacks a bit of impact. As the conclusion says: this does not (yet) fully explain acceleration. However, I think this is a good step toward it. The paper presents a promising research direction: since RK-like optimization schemes are not really popular, this may leads to new method and new kind of analysis. I recommend acceptance. Post-rebuttal ---------------- I read the rebuttal.

Reviewer 3



This paper proposed an approach to developing accelerated gradient descent methods. Following the differential equation approach in existing literature, the idea of this paper is to approximate the continuous path using Runge-Kutta discretization. The resulting algorithm can give a convergence rate arbitrarily close to O(1 / k^2), if the objective function is smooth enough. The idea is novel to the best of my knowledge, though it is perhaps direct to people who know the Runge-Kutta method. The title is too strong in my opinion. First, the optimal O(1 / k^2) convergence rate is not achieved, but only approximated. Second, how well the optimal rate can be approximated depends on how smooth the objective function is (boundedness of derivatives); this additional condition is not required in existing literature. It is interesting that now one can achieve a convergence rate better than the standard O(1 / k) via a principled approach. However, since this paper is arguably a very direct application of Runge-Kutta scheme, it does not seem clear to me that one may actually achieve the optimal O(1 / k^2) rate following this research direction. The authors should elaborate on why they believe that this work will not be obsolete in the future, in the concluding section. In Assumption 2, the objective function has to be *continuously* differentiable instead of merely differentiable, if one wants to argue for existence of an upper bound of the derivatives, by exploiting compactness of the sublevel sets. I did not have time to check the proofs. The numerical results may not be reproduced, as the synthetic dataset is not given.